# Distance- and Momentum-Based Symbolic Aggregate Approximation for Highly Imbalanced Classification

**DOI:** 10.3390/s22145095

**Published:** 2022-07-07

**Authors:** Dong-Hyuk Yang, Yong-Shin Kang

**Affiliations:** Advanced Institute of Convergence Technology, Suwon 16229, Korea; dhyang@snu.ac.kr

**Keywords:** time-series representation, symbolic aggregate approximation, momentum, highly imbalanced classification

## Abstract

Time-series representation is the most important task in time-series analysis. One of the most widely employed time-series representation method is symbolic aggregate approximation (SAX), which converts the results from piecewise aggregate approximation to a symbol sequence. SAX is a simple and effective method; however, it only focuses on the mean value of each segment in the time-series. Here, we propose a novel time-series representation method—distance- and momentum-based symbolic aggregate approximation (DM-SAX)—that can secure time-series distributions by calculating the perpendicular distance from the time-axis to each data point and consider the time-series trend by adding a momentum factor reflecting the direction of previous data points. Experimental results for 29 highly imbalanced classification problems on the UCR datasets revealed that DM-SAX affords the optimal area under the curve (AUC) among competing time-series representation methods (SAX, extreme-SAX, overlap-SAX, and distance-based SAX). We statistically verified that performance improvements resulted in significant differences in the rankings. In addition, DM-SAX yielded the optimal AUC for real-world wire cutting and crimping process dataset. Meaningful data points such as outliers could be identified in a time-series outlier detection framework via the proposed method.

## 1. Introduction

A time-series is a collection of temporal data and is one of the most frequently generated data in real-world applications. Thus, time-series analysis has been a crucial task in real-world data-mining research since time-series can be easily obtained from various data sources. To appropriately analyze a time-series, the most important task is time-series representation, which involves the extraction of feature values from the time-series. Generally, time-series consists of continuous values with enormous lengths; thus, extracting feature values that can summarize the given time-series is a crucial task.

The most widely employed approach for time-series representation is dimensionality reduction [1,2,3,4,5,6]. One of the initially used dimensionality reduction approaches is sampling [1]. In this approach, a single data point is selected for each time-series segment and is considered as the feature value that represents the corresponding segment in the time-series. Although the sampling method is easy to implement, representing each segment of the time-series involving only a single data point is difficult, particularly when there are numerous data points in each time-series segment. To improve the sampling method, extracting a feature value that can effectively represent a set of data points in each time-series segment has received significant attention. One notable method is piecewise aggregate approximation (PAA) [2], which computes the mean value of each segment in a time-series to represent the corresponding set of data points. PAA has been demonstrated to be effective for time-series representations. Consequently, various extensions have been introduced in time-series representations [3,4,5,6].

Another broadly employed approach to represent time-series is discretization, which converts the numeric value to a symbolic form [7,8,9,10,11,12]. Specifically, this method discretizes the time-series into a predefined number of segments and then converts each segment into a symbol. One of the widely used time-series discretization methods is symbolic aggregate approximation (SAX) [11], which transforms the results from PAA values to a symbol sequence. The time-series distribution space that follows the standard normal distribution was divided into equiprobable regions. Each region is represented by a specific symbol, such that each segment can be mapped into a corresponding symbol where it exists. SAX easily allows inspection of results using discretized symbols in real-world applications [13,14,15,16,17,18,19,20,21,22,23,24,25,26].

Nevertheless, SAX has a major limitation in which it only represents the mean value of each segment in the time-series. Thus, SAX representation is prone to missing some important information in the time-series [27,28,29,30,31,32,33,34,35,36,37,38,39,40,41]. Especially, in classification, one of the main research topics in time-series analysis, retention of meaningful information is critical because the classification performance would be significantly affected if the symbols between different classes are ambiguously discriminated. Moreover, generating symbols that can properly represent the corresponding class is a key consideration in a highly imbalanced classification, where the number of data points between different classes is extremely different. By employing conventional SAX, the segment that contains data points of the minor class might be converted to a symbol that does not reflect them because of the relatively larger number of data points corresponding to the major class. Thus, the influence of data points in the minor class would be diminished during time-series representation. In fact, dealing with highly imbalanced data is one of the main characteristics of real-world applications [42,43,44,45]. Therefore, a time-series discretization method that can effectively summarize data points to properly represent the class which they reside in must be developed.

Herein, we propose a novel time-series representation method, named distance- and momentum-based symbolic aggregate approximation (DM-SAX), that can discriminate between majority and minority classes by considering time-series distributions and trends. As demonstrated in later sections, the proposed method considers the time-series distribution by calculating the perpendicular distance from the time-axis to each data point. In addition, the time-series trend is considered by adding a momentum factor that reflects the direction of previous data points. It will be easy to identify the meaningful data points by employing DM-SAX, such as defects in manufacturing process, in a time-series outlier detection framework.

The remainder of this paper is organized as follows. Section 2 reviews the related works. In Section 3, the conventional SAX and the proposed DM-SAX are introduced in detail. Section 4 presents the performance benchmarks of the proposed model against other time-series symbolic-representation approaches. Finally, the conclusions and possible avenues for future research are presented in Section 5.

## 2. Related Works

### 2.1. Conventional SAX

The conventional SAX represents and preserves time-series information using alphabetical symbols. It is well known for its effective representation of high-dimensional time- series while maintaining the properties of the given data points in the time-series [11].

Figure 1 presents the SAX procedure. The first phase is to employ dimensionality reduction using PAA [2]. As shown in Figure 1a, the time-series is divided into segments with a certain length, and each segment is summarized with the mean value of the data points that it includes. Therefore, the time-series vector $X=\left[x_{1},\;\ldots ,\;x_{N}\right]$ with a length of *N* is converted into a PAA vector $X^{PAA}=\left[x_{1}^{PAA},\;\ldots ,\;x_{S}^{PAA}\right]$ with a length of *S*. The *i*th element of PAA $x_{i}^{PAA}$ is computed using the equation below.
(1)$$x_{i}^{PAA}=\frac{S}{N}\overset{\left(\frac{N}{S}\right)i}{\underset{j=\left(\frac{N}{S}\right)\left(i-1\right)+1}{\sum }}x_{j},$$
where *i* ranges from 1 to *S*, and $x_{j}\;$is the *j*th element of X.

Here, a constant $\frac{N}{S}$ is called the *time segment size* (*t_size*), which is used as the main PAA hyperparameter.

The second phase involves discretizing the PAA values, as shown in Figure 1b. In this phase, a previously generated PAA vector $X^{PAA}=\left[x_{1}^{PAA},\;\ldots ,\;x_{S}^{PAA}\right]$ is transformed into a symbol vector $X^{SAX}=\left[x_{1}^{SAX},\;\ldots ,\;x_{S}^{SAX}\right]$ by mapping each element of $X^{SAX}$ into one of the discretization regions in accordance with its value. Note that the discretization regions follow a standard normal distribution, with the size of each region being equal to satisfy the equiprobability. For instance, Figure 1b demonstrates a case with an alphabet size of 7, indicating that ±1.07, ±0.57, and ±0.18 are the ‘breakpoints’ of each separation, and that each alphabet (a, b, c, d, e, f, and g), following the standard normal distribution, occupies 14.3% of the area. Table 1 lists the breakpoints.

Finally, the element is converted to an alphabetical symbol, becoming the represented value for its corresponding element of $X^{SAX}$. At this point, the number of discretization regions is called the *number of bins* (*n_bins*), which is employed as the main SAX hyperparameter.

### 2.2. Real-World Applications of SAX

SAX is a popular time-series representation method that has been extensively studied in real-world applications. In general, there are two research topics related to SAX in real-world applications, such as pattern-discovery and prediction.

With this, the pattern-discovery is an interesting research topic. Park and Jung [13] proposed a pattern-discovery framework that combined SAX with association rule mining (ARM). In the SAX-ARM method, time-series generated from sensors in a die-casting process are converted to symbols. Then, apriori, one of the most employed ARM algorithms, extracts the deviant patterns from those symbols. Ferreira et al. [14] suggested adaptive SAX (ASAX) to analyze heat-wave patterns from daily information. The suggested approach adopts SAX after time-series segments are automatically adapted by considering the difference between the current and average values. Similarly, Wu and Lee [15] introduced an algorithm called closed flexible patterns (CFP) to identify the mining of closed flexible patterns by utilizing SAX. CFP employs SAX to convert time-series into symbols. Subsequently, frequent patterns are extracted through a depth-first search. Ohsaki et al. [16] suggested a rule discovery support system for sequential medical data. The proposed system utilizes SAX to extract patterns of glutamic pyruvic transaminase (GPT) from data obtained from patients with hepatitis. In the medical field, Tseng et al. [17] proposed a SAX modification to identify novel genetic relationships by mining similar subsequences in microarray data. Ordóñez [18] proposed a novel pattern-visualization algorithm that can differentiate between medical conditions such as renal and respiratory failure. The proposed algorithm applies SAX to help interpret time-series data obtained from the pediatric intensive care unit (PICU). Yaik et al. [19] employed SAX to identify frequent patterns generated in CPU traces. By using SAX, the proposed method can predict longer steps ahead than the conventional prediction technique (i.e., network weather services (NWS)).

Another research topic is prediction. Pouget et al. [20] suggested an approach that can detect attacks that occurred on the Internet. This approach uses SAX to transform data collected in a honeypot platform into symbols, which are used to detect attacks by systematically identifying similarities between the time signatures of the attack tools. On the other hand, Zoumboulakis and Roussos [21] proposed a novel method to detect complex events in sensor networks. Here, the real-valued sensor data are converted to symbols via SAX representation, and complex events that are difficult or impossible to describe using conventional SQL-like languages are detected using distance metrics. Meanwhile, McGovern et al. [22] introduced a prediction system that can detect severe weather conditions such as tornados. The introduced system applies SAX to convert large multidimensional time-series into symbolic representations. Symbols that satisfy the predefined probability of detection (POD) and false alarm ratio (FAR) are selected to create rules that can identify tornados. Ciompi et al. [23] adopted a technique for the automatic detection of diseased regions of vessels using intravascular ultrasound (IVUS) sequences. Morphological profiles from IVUS were obtained using the proposed technique. Thereafter, SAX was applied to convert morphological profiles to discrete codewords, which were used in the selection of keyframes that can detect unhealthy regions of the vessel. Shie et al. [24] proposed an online treatment system for panic patients by combining biofeedback therapy and web technologies. Numerical biofeedback data are transformed into symbolized sequence data by employing SAX, and the classify-by-sequence (CBS) algorithm is applied to detect whether the treatment is suitable. Morgan et al. [25] proposed an anomaly detection algorithm for marine engines. Here, the measured iron concentrations from the cylinder of the engine are collected. Then, these measurements are converted to symbols by applying SAX, and support vector machines (SVM) are employed to detect unexpected concentrations in the engine. He et al. [26] proposed an analog circuit fault detection system using SAX. In the proposed system, data are collected from four op-amp bi-quad low-pass filter circuits and then converted to symbols to detect the type of fault.

### 2.3. Variations of SAX

SAX results in an appropriate time-series representation. However, as previously discussed, SAX is based on the PAA representation. Therefore, it only symbolizes the mean value of each segment in the time-series, and this representation might cause information loss. Various attempts have been devoted toward overcoming the shortcomings in existing literature.

Fuad and Marwan [27] proposed extreme-SAX (E-SAX), where the symbols can represent the segment more precisely than those of conventional SAX by considering only the minimum and maximum data points of the segment. Lkhagva et al. [28] used extended-SAX to reflect the trend of time-series containing a few critical data points, such as financial time-series. The proposed approach can offset the negative effect and only consider the mean value of the segment by adding the minimum and maximum values to the mean value. Lin et al. [29] proposed bag-of-patterns (BOP), which constructs a histogram of SAX words using the framework of bag-of-feature (BOF). Thereafter, classification is performed by comparing the histograms to identify the nearest neighbor located in the training set. One of the popular variations of BOP is symbolic aggregate approximation and vector space model (SAX-VSM) [30], which introduces term frequency-inverse document frequency (TF-IDF) to assign weights to SAX words. Each SAX word has a different weight for each class to optimize the similarity computation to a certain extent. The major contribution of SAX-VSM is the proposal of a parameter selection optimization method, DIRECT, to accelerate the SAX parameter search. Fuad and Marwan [31] suggested overlap-SAX (O-SAX) to include the trend information of a given time-series. The last data point in the previous segment and the first data point in the following segment are swapped to consider the trend of the data points. Song et al. [32] proposed a novel approach referred to as transitional-SAX (T-SAX) to incorporate transitional information into conventional SAX. To retain meaningful information, the proposed approach retains the upward and downward transitional information by tracing the data points traveling from the current quantile region to the next location. Sun et al. [33] suggested SAX-based trend distance (SAX-TD) to reflect the trend of the time-series using the first and last data points of a segment. Yin et al. [34] proposed the trend feature symbolic approximation (TFSA) to enhance the classification performance of SAX. In the proposed approach, a two-stage segmentation approach for fast segmentation of long time-series is applied, and the experimental results demonstrate that it achieves better segmentation and classification accuracy than SAX. Malinowski et al. [35] adopted a novel algorithm 1d-SAX that outperformed SAX, while retaining the compression ratio. In the algorithm, linear regression is applied in sub-segments of the time-series. Then, symbols are created via mean and slope values. Fuad and Marwan [36] proposed the genetic algorithm SAX (GASAX) to determine the breakpoints using a genetic algorithm. In the proposed algorithm, a genetic algorithm is employed to determine the nearly optimal configuration of breakpoints that provides the optimal fitness during the SAX process. Additional variations of SAX are described in [37,38,39,40,41].

## 3. Proposed Method: DM-SAX

### 3.1. D-SAX

The conventional SAX approach results in an appropriate time-series representation. However, SAX is based on the PAA representation, minimizing the dimensionality by calculating the mean values of equal-sized segments. This implies that the mean value-based representation might overlook some important values in industrial time-series, such as outliers. In this section, we propose a two-stage time-series representation method that can summarize the time-series better than the conventional SAX algorithm.

The first stage of representing time-series in the proposed method involves the consideration of the distribution of the time-series by computing the perpendicular distance from the time-axis to each data point in the segment. It should be noted that the perpendicular distance from the time-axis to the data point implies the absolute value of the data point. For instance, the 2nd value in Figure 2 is −2; hence, the perpendicular distance from the time-axis to −2 is 2. By considering the distribution of the time-series, the information with important data points such as outliers can be preserved. Therefore, the time-series vector $X=\left[x_{1},\;\ldots ,\;x_{N}\right]$ with length *N* is converted into a distance-based PAA (D-PAA) vector $X^{D-PAA}=\left[x_{1}^{D-PAA},\;\ldots ,\;x_{S}^{D-PAA}\right]$ with length *S*. The *i*th element of D-PAA $x_{i}^{D-PAA}$ is expressed as,
(2)$$x_{i}^{D-PAA}=\overset{\left(\frac{N}{S}\right)i}{\underset{j=\left(\frac{N}{S}\right)\left(i-1\right)+1}{\sum }}x_{j}\;\frac{\left|x_{j}\right|}{\sum_{j=\left(\frac{N}{S}\right)\left(i-1\right)+1}^{\left(\frac{N}{S}\right)i}\left|x_{j}\right|}\;,\;$$
where *i* ranges from 1 to *S*, $x_{j}\;$is the *j*th element of X, and $\left|x_{j}\right|$ is the absolute value of the *j*th element of X.

Afterward, a previously generated D-PAA vector $X^{D-PAA}=\left[x_{1}^{D-PAA},\;\ldots ,\;x_{S}^{D-PAA}\right]$ is converted into a symbol vector $X^{D-SAX}=\left[x_{1}^{D-SAX},\;\ldots ,\;x_{S}^{D-SAX}\right]$. In this phase, the same discretization and symbolization are processed similar to that of the SAX. In this study, we refer to this method as distance-based SAX (D-SAX). The process of D-SAX is shown in Figure 2.

### 3.2. DM-SAX

Although considering the distribution of data points in a time-series is an effective method, this method does not reflect the trend of the time-series. Data points in the first segment shows an increasing trend while data points in the second segment show a decreasing trend, as shown in Figure 3a. Considering only the distribution of data points by calculating the perpendicular distance from the time-axis to the data points is not sufficient to appropriately represent a given time-series.

The second stage to represent a time-series in the proposed method involves adding a momentum factor to consider the time-series trend. The equation for the momentum factor is,
(3)$$m_{t}=am_{t-1}+\eta \left(x_{t}-x_{t-1}\right)$$
where $t=\left(\frac{N}{S}\right)i,$ $m_{\left(\frac{N}{S}\right)\left(i-1\right)+1\;}=0,$ and $x_{t}\;$is the *t*th element of X.

Note that *a* is the hyperparameter reflecting the direction of previous data points, and η is the hyperparameter controlling the gradient of the current and previous data points. 

The trend of data points is effectively reflected by considering the trend of the time-series via the momentum factor that can reflect the direction of the time-series. After adding the momentum factor to the D-PAA process, the time-series vector $X=\left[x_{1},\;\ldots ,\;x_{N}\right]$ with length *N* is converted into a distance- and momentum-based PAA (DM-PAA) vector $X^{DM-PAA}=\left[x_{1}^{DM-PAA},\;\ldots ,\;x_{S}^{DM-PAA}\right]$ with length *S*. Finally, the *i*th element of DM-PAA $x_{i}^{DM-PAA}$ is given by,
(4)$$x_{i}^{DM-PAA}=x_{i}^{D-PAA}+m_{t}$$

Note that, when *a* and η are 0, the result of DM-PAA is the same as that of D-PAA.

Then, a previously generated DM-PAA vector $X^{DM-PAA}=\left[x_{1}^{DM-PAA},\;\ldots ,\;x_{S}^{DM-PAA}\right]$ is converted into a symbol vector $X^{DM-SAX}=\left[x_{1}^{DM-SAX},\;\ldots ,\;x_{S}^{DM-SAX}\right]$. In this phase, the same discretization and symbolization are processed in the same manner as in the SAX. In this study, we refer to this method as DM-SAX. Figure 3 shows the process of DM-SAX.

## 4. Experimental Validation

In this section, we experimentally evaluated whether the proposed DM-SAX is superior to other methods on various datasets provided by the University of California—Riverside (UCR) time-series classification archive [46], a well-known data repository for time-series data mining research, and real-world manufacturing processes.

### 4.1. UCR Datasets

#### 4.1.1. Experimental Design

The comparative classification performances of five time-series representation methods (SAX, extreme-SAX (E-SAX), overlap-SAX (O-SAX), D-SAX, and DM-SAX) are presented on 29 different highly imbalanced datasets taken from the UCR time-series classification archive. This archive originally contained 128 datasets involving various numbers of data points, input features, and classes. For highly imbalanced classification, which is the scope of our study, we converted the class with the smallest number of data points to a positive class, whereas the other classes were converted to a negative class. Then, we calculated the imbalance ratio for each dataset (i.e., the proportion of the number of data points in the negative class to the number of data points in the positive class). Afterward, datasets with imbalance ratios greater than 10 were selected for this experiment, reducing the number of datasets from 128 to 29. Note that the datasets were originally divided into training and test set. Table 2 lists the datasets used.

The experiment was controlled such that a random forest with 20 iterations was used as a base classifier since it is well known for its stable predictive performances [47,48,49,50]. As previously discussed, *t_size* and *n_bins* are the two main hyperparameters of the SAX. In this experiment, we set *t_size* to 3 and 5 and *n_bins* to 4, 6, 8, and 10; thus, a total of 8 experiments were conducted. Note that classes containing a positive class are represented as positive classes in the PAA process. For example, classes 0, 1, 0, 0, 0, and 0 are converted to 1 and 0 if *t_size* is set to 3. For DM-SAX, *a* and η were fixed at 0.9 and 0.01, respectively. Note that the area under the curve (AUC) was employed as a performance measure because it is regarded as a comprehensive and balanced metric that better reflects the classification performance on highly imbalanced data [51,52].

#### 4.1.2. Experimental Results

Table 3 summarizes the results of the performance benchmarks. The AUCs were obtained by averaging the results from the validation repeated eight times, as mentioned above. The highest AUCs obtained for each dataset are highlighted in bold. On an average, DM-SAX achieved the highest AUC, 73.44(%), followed by D-SAX, E-SAX, O-SAX, and SAX. Moreover, DM-SAX demonstrated an optimal performance with a mean rank value of 2.24. Specifically, DM-SAX outperformed the other methods in 10 out of 29 datasets. Furthermore, we recognized that considering both the distribution and trend of the time-series resulted in a more beneficial effect than solely considering the distribution of the time-series in 16 out of 29 datasets.

Note that DM-SAX was superior to conventional SAX particularly when the dataset was difficult to classify, with *DistalPhalanxTW*, *MiddlePhalanxTW*, *Phoneme*, *PigArtPressure*, and *ProximalPhalanxTW* being the typical cases in point. It may be hard to attribute these comparative results to a specific factor. Nevertheless, the results indicate that time-series representation by calculating the perpendicular distance from the time-axis to each data point and computing the trend of data points resulted in data representation that could appropriately deal with ‘hard-to-classify’ problems.

The Friedman omnibus test [53] was first performed on the rank values of the classification performances for each competing method across the datasets to verify the statistical significance of the difference between the methods. Therefore, the *p*-value (<0.5 × 10^−4^) was demonstrated to be less than the alpha risk of 0.05, indicating statistically significant differences in the rankings between the AUCs of time-series representation methods. Subsequently, a post-hoc Wilcoxon rank test was employed to enforce the pairwise comparison of the time-series representation methods, with an adjusted alpha risk of 0.005 (=0.05/10) [54,55].

Table 4 presents the test results. Although there was no statistically significant difference between DM-SAX and D-SAX, DM-SAX outperformed SAX, E-SAX, and O-SAX, whereas D-SAX was observed to be insignificant in contrast to DM-SAX. This indicates that the computation of the time-series trend redeemed the classification performance of the method that only considered the distribution of the time-series.

Figure 4 shows the ratio of each algorithm included in the top-n rank by AUC. DM-SAX is considered the top-performing algorithm in 31% (73/232) of repeated experiments among 29 datasets, and it was at least the 2nd ranked algorithm in 59% (137/232) of the results. Overall, DM-SAX showed a better classification performance than the other methods.

### 4.2. Real-World Manufacturing Process Dataset

#### 4.2.1. Experimental Design

A manufacturing process dataset compiled from cutting and crimping process in the wiring harness manufacturing was used to further prove the applicability of the proposed DM-SAX. A wiring harness is used to transmit electrical signals between control devices in a vehicle. To produce a wiring harness, a cutting machine was used to cut the wire to a certain length. Then, both ends of the wire were connected to the terminals and were pressed using an applicator.

The dataset was collected from 20:38 19 July to 13:02 22 July 2021, with 285,297 data points, and each consecutive 100 data points represented approximately 1 min. Failures were recorded at 656 data points, and the imbalance ratio was 433, indicating a highly imbalanced ratio. In this section, three features (B/S, RCFA, and MPP) are used to predict whether the products prepared by wire cutting and crimping are normal or abnormal. Table 5 and Table 6 lists a brief description and detailed statistical information on these features, respectively.

There were two major differences although the overall experimental design was almost the same as that of the UCR datasets. One major difference is the training and test split criterion. As previously mentioned, training and test sets were originally divided in UCR datasets. In contrast, we arbitrarily divided the real-world dataset into training and test sets in a ratio of 0.7 and 0.3. The other difference is that we set *t_size* to 25, 50, 75, 100, and 150 for the real-world dataset, which is larger than those on the UCR dataset experiments. Table 7 summarizes the detailed similarities and differences between the experiments on the real-world and UCR datasets.

#### 4.2.2. Experimental Results

Table 8 lists the experimental results, and the best AUCs for each case are marked in bold. The results demonstrate that DM-SAX obtained the optimal AUC (98.88%), followed by D-SAX, E-SAX, O-SAX, and SAX. In addition, DM-SAX demonstrated the optimal performance while outperforming other methods in 10 out of 20 experimental cases, with a mean rank value of 1.15.

Note that DM-SAX outperformed D-SAX, particularly when the *t_size* was larger than 100. This implies that the addition of a momentum factor resulted in a favorable effect when there were sufficient data points to reflect the overall trend of the time-series.

## 5. Conclusions

In this study, we developed a novel time-dimensionality representation method, called DM-SAX, and compared it with other well-known time-series representation methods. The proposed method secures the time-series characteristics by computing the perpendicular distance from the time-axis to data points and considers the trend of time-series by employing the momentum factor that can reflect the direction of previous data points.

The experimental results on 29 UCR problems proved that DM-SAX exhibited the optimum AUC among the competing methods. Moreover, we empirically verified that DM-SAX is superior to other methods using real-world wire cutting and crimping process data. Defect detection would be applicable in the real-time industrial process using the proposed method. To be more specific, if the symbols generated in the proposed method are located at both ends of the discretization region, one could easily determine that those symbols represent the defects. Furthermore, the proposed method can also be employed in unsupervised learning, such as for human behavior pattern discovery, traffic pattern discovery, and failure rule discovery.

As an extension of the proposed method, a new type of factor that can further represent the characteristics of a given time-series will be developed in the future. Here, an additional factor related to momentum factor that could better reflect the trend of the data will be considered. In addition, a heuristic method for selecting *a* and η may be another future research topic. The current configuration (*a*: 0.9, η: 0.01) may have overlooked the optimal trend of the time-series. Thus, it is necessary investigating various search algorithms.

## Figures and Tables

**Figure 1 sensors-22-05095-f001:**
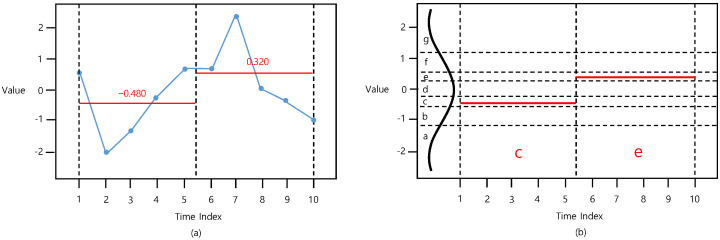
Procedure of (**a**) PAA (*t_size* = 5) and (**b**) SAX (*n_bins* = 7).

**Figure 2 sensors-22-05095-f002:**
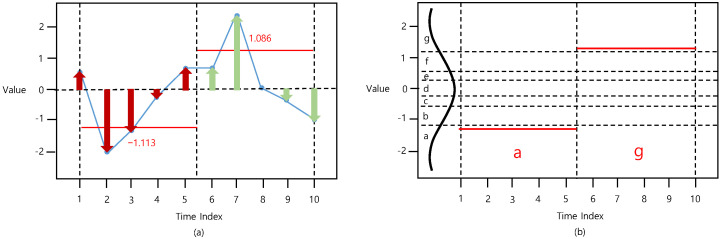
Procedure of (**a**) D-PAA (*t_size* = 5) and (**b**) D-SAX (*n_bins* = 7).

**Figure 3 sensors-22-05095-f003:**
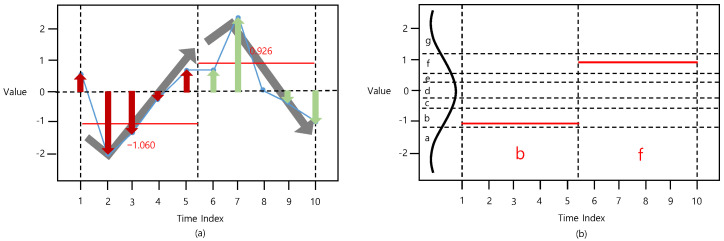
Procedure of (**a**) DM-PAA (*t_size* = 5) and (**b**) DM-SAX (*n_bins* = 7).

**Figure 4 sensors-22-05095-f004:**
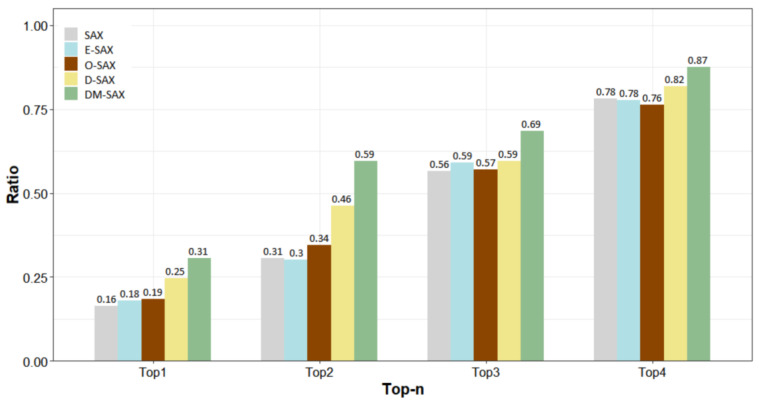
Ratio of each algorithm included in the top-*n* rank on 29 UCR datasets.

**Table 1 sensors-22-05095-t001:** Lookup table containing the breakpoints.

	* **n_bins** *	3	4	5	6	7	8	9	10
** *β_i_* **									
$\beta_{1}$		−0.43	−0.67	−0.84	−0.97	−1.07	−1.15	−1.22	−1.28
$\beta_{2}$		0.43	0.00	−0.25	−0.43	−0.57	−0.67	−0.76	−0.84
$\beta_{3}$			0.67	0.25	0.00	−0.18	−0.32	−0.43	−0.52
$\beta_{4}$				0.84	0.43	0.18	0.00	−0.14	−0.25
$\beta_{5}$					0.97	0.57	0.32	0.14	−0.00
$\beta_{6}$						1.07	0.67	0.43	0.25
$\beta_{7}$							1.15	0.76	0.52
$\beta_{8}$								1.22	0.84
$\beta_{9}$									1.28

**Table 2 sensors-22-05095-t002:** Dataset descriptions.

Dataset	#Training Data Points	#Test Data Points	#Input Features	Imbalance Ratio
Adiac	390	391	176	38.1
CricketX	390	390	300	11.0
CricketY	390	390	300	11.0
CricketZ	390	390	300	11.0
Crop	7200	16,800	46	23.0
DistalPhalanxOutlineAgeGroup	400	139	80	11.0
DistalPhalanxTW	400	139	80	19.7
ECG5000	500	5000	140	207.3
ElectricDevices	8926	7711	96	12.3
EOGHorizontalSignal	362	362	1250	11.3
EOGVerticalSignal	362	362	1250	11.3
FaceAll	560	1690	131	45.9
FacesUCR	200	2050	131	45.9
FiftyWords	450	455	270	149.8
Fungi	18	186	201	24.5
InsectWingbeatSound	220	1980	256	10.0
MedicalImages	381	760	99	48.6
MiddlePhalanxTW	399	154	80	15.3
NonInvasiveFetalECGThorax1	1800	1965	750	49.2
NonInvasiveFetalECGThorax2	1800	1965	750	49.2
OSULeaf	200	242	427	10.6
Phoneme	214	1896	1024	1054.0
PigAirwayPressure	104	208	2000	51.0
PigArtPressure	104	208	2000	51.0
PigCVP	104	208	2000	51.0
ProximalPhalanxTW	400	205	80	32.6
ShapesAll	600	600	512	59.0
SwedishLeaf	500	625	128	14.0
WordSynonyms	267	638	270	74.4

**Table 3 sensors-22-05095-t003:** Performance benchmarks (UCR datasets).

Dataset	SAX	E-SAX	O-SAX	D-SAX	DM-SAX
Adiac	43.10	**52.64**	50.01	48.34	48.97
CricketX	55.78	58.68	60.93	**61.01**	60.23
CricketY	68.84	71.62	63.95	**72.73**	72.54
CricketZ	51.33	**52.90**	51.57	52.42	52.86
Crop	99.55	99.42	**99.73**	99.64	99.64
DistalPhalanxOutlineAgeGroup	89.31	**93.06**	83.92	82.48	81.54
DistalPhalanxTW	50.73	54.51	57.16	56.74	**57.51**
ECG5000	65.72	58.09	60.87	66.54	**67.04**
ElectricDevices	80.55	78.34	80.56	**84.80**	84.02
EOGHorizontalSignal	72.34	76.00	**77.95**	73.75	74.37
EOGVerticalSignal	69.57	70.84	**76.62**	69.94	69.14
FaceAll	94.29	92.89	89.28	96.09	**97.15**
FacesUCR	61.33	55.96	59.05	**65.01**	63.88
FiftyWords	60.54	**65.93**	58.70	60.86	61.54
Fungi	**98.12**	86.14	93.89	97.77	97.89
InsectWingbeatSound	76.53	62.60	71.30	78.31	**78.83**
MedicalImages	77.95	87.85	90.21	95.24	**95.75**
MiddlePhalanxTW	63.70	69.12	68.31	66.15	**71.17**
NonInvasiveFetalECGThorax1	85.80	87.12	67.86	**92.76**	92.31
NonInvasiveFetalECGThorax2	82.04	81.11	67.12	87.23	**87.72**
OSULeaf	57.59	47.28	56.51	57.78	**58.30**
Phoneme	36.33	69.76	**69.82**	53.56	53.52
PigAirwayPressure	59.11	81.37	**84.45**	64.83	66.59
PigArtPressure	60.70	51.64	39.82	76.63	**77.15**
PigCVP	73.18	47.11	60.54	**86.58**	84.24
ProximalPhalanxTW	56.00	72.25	55.87	71.74	**74.06**
ShapesAll	82.38	74.49	**89.97**	85.76	84.61
SwedishLeaf	64.70	**72.39**	59.80	65.72	63.80
WordSynonyms	51.43	**57.13**	57.09	54.18	53.32
Mean AUC (%)	68.57	69.94	69.06	73.26	**73.44**
Mean Rank	3.86	3.21	3.24	2.41	**2.24**

**Table 4 sensors-22-05095-t004:** Post-hoc test (Wilcoxon) results (*p*-value).

	SAX	E-SAX	O-SAX	D-SAX	DM-SAX
SAX	-	0.9573	0.6517	0.0135	0.0022
E-SAX		-	0.9222	0.0139	0.0032
O-SAX				0.0251	0.0043
D-SAX				-	0.2692
DM-SAX					-

**Table 5 sensors-22-05095-t005:** Description of features.

Features	Description
B/S	Bad limit overall/Specification delta conductor
RCFA	Results measured from crimp force analyzer
MPP	Maximum press power

**Table 6 sensors-22-05095-t006:** Descriptive statistics.

Features	Min	Median	Mean	Max
B/S	−2052.0	1.0	−1.1	1674.0
RCFA	1.0	14.0	17.4	2052.0
MPP	99.0	3457.0	3774.8	8758.0

**Table 7 sensors-22-05095-t007:** Similarities and differences between experiments of UCR and real-world datasets.

	Elements	UCR	Real-World
Similarities	Competing methods	SAX, E-SAX, O-SAX, D-SAX, and DM-SAX	
	Performance measure	AUC	
	Base classifier	Random forest (20 iterations)	
	*n_bins*	4, 6, 8, and 10	
	*a*	0.9	
	η	0.01	
Differences	*t_size*	3, 5	25, 50, 75, 100, and 150
	Training/Test set ratio	Originally split in the archive	0.7/0.3

**Table 8 sensors-22-05095-t008:** Performance benchmarks (real-world dataset).

*t_size*	*n_bins*	SAX	E-SAX	O-SAX	D-SAX	DM-SAX
25	4	85.16	89.45	87.74	**99.39**	99.38
	6	85.00	94.18	93.58	99.73	**99.76**
	8	80.30	93.95	93.69	**99.88**	**99.88**
	10	83.23	93.68	92.28	**99.34**	99.33
50	4	85.90	81.48	86.89	**98.95**	**98.95**
	6	82.32	89.86	89.51	**99.02**	**99.02**
	8	84.67	93.17	87.87	**99.50**	99.47
	10	84.72	93.39	90.14	**99.52**	**99.52**
75	4	79.29	77.62	85.28	**99.27**	**99.27**
	6	76.83	88.39	86.97	**98.96**	**98.96**
	8	78.56	91.91	86.79	**98.57**	**98.57**
	10	76.23	92.53	86.79	98.95	**98.96**
100	4	84.58	76.34	85.81	98.39	**98.40**
	6	81.79	89.40	86.52	98.55	**98.56**
	8	78.48	92.79	87.77	97.79	**98.27**
	10	76.17	93.57	86.44	98.44	**98.91**
150	4	77.27	73.90	82.60	97.75	**97.76**
	6	78.45	86.71	82.03	98.06	**98.08**
	8	82.87	91.48	80.41	98.62	**98.66**
	10	79.05	91.37	78.79	97.45	**97.90**
Mean AUC (%)		81.04	88.76	86.90	98.81	**98.88**
Mean Rank		4.70	3.40	3.90	1.50	**1.15**

## Data Availability

Not applicable.

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
