# Peer review of "Distance- and Momentum-Based Symbolic Aggregate Approximation for Highly Imbalanced Classification"

_sensors, 2022, doi:10.3390/s22145095_

Round 1

Reviewer 1 Report

The paper proposes an improved version of the SAX method for converting time series in a symbolic representation. The improved version uses a modified expression for the mean and an additional term quantifying the local trend of the time series. The efficiency of the method is tested in case of time series classification, on a large set of imbalanced datasets taken from the UCR archive. As summarized in Table 2, the proposed method (called DM-SAX) provides improved results. DM-SAX obtains the best classification in 10 out of 29 cases while the E-SAX method (also an improved version of SAX, proposed by Fuad and Marwan in 2020) provides the best results for 6 out of 29 cases. DM-SAX loses the competition with a difference higher than 5% in 5 cases while the same happens for E-SAX 13 cases.

In my opinion the paper is in general well written and the results worth to be published. The idea of the paper could be useful in various applications.

 I would recommend the authors to take into account the following remarks:

-          The paper contains an exhaustive review on the SAX method (maybe a bit too long) but it misses a brief description of the E-SAX and O-SAX methods which are used for comparison.

-          The sub-section 4.2 is dedicated to a real-life case. The case if interesting but, however, my impression is that it was selected mainly because in this particular case the DM-SAX method obtains good results in comparison with the other methods. In my opinion it does not serve the case of the proposed method, the comparison in Table 2, which is unbiased put it a a better light.

-          I don’t see which is in fact the difference between “the perpendicular distance 223 from the time-axis to each data point in the segment” the value of the time series.

-          Figs 1-3 caption should be extended to include an explanation helping the reader to understand the figures (besides the explanation existing in the text)

-          On page 10, Fig. 3 is in fact Fog. 4. Also in the last paragraph on this page I think that “Fig. 2 illustrated …” should be replaced by ‘Fig. 4 illustrates …’

Reviewer 2 Report

Review of "Distance and Momentum based Symbolic Aggregate Approximation for Highly Imbalanced Classification" by Yang & Kang (2022)

Authors proposed symbolic aggregate approximation (SAX) index to represent time-series, which converts the results from piecewise aggregate approximation (PAA) to a symbol sequence. Specifically, they proposed the distance and momentum-based symbolic aggregate approximation (DM-SAX). Method was employed in 29 highly imbalanced classification problems on the UCR datasets to show the results. I think that paper could be considered for publication in Sensors journal, after addressing the following issues:

1. L49: add in "[7-11]" the reference Zhao et al. (2017) where is considered the discretization of time series for entropy computation.

2. L198-199: Put this in a whole line.

3. L201: Authors could add a sentence about approach of Eq. (1) is related to Detrended Cross-Correlation Analysis (DCCA) for block divisions of time series (Contreras-Reyes & Idrovo-Aguirre, 2020).

4. L227: "D-PAA" <-> "DPAA" (and in all manuscript).

5. L233-235: Put this in a whole line.

6. L240: "D-SAX" <-> "DSAX" (and in all manuscript).

7. Eq. (3): replace by the eta greek letter. Also, define eta parameter.

8. L253-255: Put this in a whole line.

9. L263: "DM-PAA" <-> "DMPAA" (an in all manuscript).

10. L283: put here what means UCR.

References:

Zhao, X., Shang, P., Huang, J. (2017). Mutual-information matrix analysis for nonlinear interactions of multivariate time series. Nonlinear Dynamics 88(1), 477-487.

Contreras-Reyes, J.E., Idrovo-Aguirre, B.J. (2020). Backcasting and forecasting time series using detrended cross-correlation analysis. Physica A 560, 125109.

Reviewer 3 Report

This paper proposes a new time-series representation method, named DM-SAX. The article proposes an interesting novel procedure and includes classification results with both literature datasets and an interesting application. However, I have a few minor remarks.

- The issue of time series representation, which is crucial in this paper, is not very well explained. Readers that are not familiar with such a concept can find hard to understand sentences like "To appropriately analyze a time-series process, the most important task is to represent a given time series. The most widely employed approach in time-series representation is dimensionality reduction". What do the authors mean by representation? Why is it useful in time series context? I think that the material in Section 2.1 can be placed in the introduction to improve the presentation of the problem.

- I have some comments about the paper structure. For example, Section 2.2 briefly discusses some variations of the SAX without showing the SAX, which is presented in Section 3.1. I suggest considering a change in the article outline: moving the material in Section 2.1 in the Introduction; discussing the SAX with its extensions (ie. merging Section 3.1 with Section 2.2) and, then, discussing the novel methodological proposal.

- How SAX and the DM-SAX can be used for unsupervised learning of time series data? Some sentences can be provided in the conclusions.
